# LEARNING TO REASON AS ACTION ABSTRACTIONS WITH SCALABLE MID-TRAINING RL

**Shenao Zhang**[1,*]**, Donghan Yu**[2]**, Yihao Feng**[2]**, Bowen Jin**[3,*]**, Zhaoran Wang**[1]**,**
**John Peebles**[*,†]**, Zirui Wang**[2,†]
[1]Northwestern University, [2]Apple, [3]UIUC

## ABSTRACT

Large language models excel with reinforcement learning (RL), but fully unlocking this potential requires a mid-training stage. Intuitively, an effective mid-training stage should both learn a strong policy prior and enable fast learning through online interactions. We formalize this intuition by presenting the first theoretical result on how mid-training shapes post-training: it acquires strong policy priors by efficiently pruning the action space and accelerates RL convergence by shortening the effective planning horizon. Moreover, we prove that temporal abstractions simultaneously compress the size of the action set and reduce the decision horizon, thereby improving regret minimization after training. Building on these insights, we introduce Reasoning as Action Abstractions (RA3), a scalable mid-training algorithm. Specifically, we derive a temporal variational bound and optimize it by iteratively discovering temporally-consistent latent structures via RL, then fine-tuning on the bootstrapped data. Experiments on code generation tasks demonstrate the effectiveness of our approach. Across multiple base models, RA3 improves the average performance on HumanEval and MBPP by 8 and 4 points over the base model and the next-token prediction baseline. Furthermore, RA3 achieves faster convergence and higher asymptotic performance in RLVR on HumanEval+, MBPP+, LiveCodeBench, and Codeforces.

## 1 INTRODUCTION

The potential of reinforcement learning (RL) as a universal policy-improvement operator has been demonstrated with remarkable success in training large language models (LLMs), spanning applications in preference optimization (Ouyang et al., 2022), mathematics (Guo et al., 2025; Zeng et al., 2025b), code generation (Yang et al., 2025; Zeng et al., 2025a), and agentic tasks (Team et al., 2025; Zhou et al., 2025b). A key factor behind these successes is the strengthened policy prior, typically obtained through mid-training (Wang et al., 2025c; Su et al., 2025), which is continued pre-training on expert data sampled from the optimal policy. Despite its widespread use, the precise role of mid-training in shaping post-training RL remains poorly understood. Without such understanding, it is difficult to design principled and effective mid-training algorithms. Heuristic metrics, such as the performance or entropy of the initial RL policy, provide only indirect signals and do not by themselves guarantee improved downstream performance.

In this paper, we propose the first theoretical analysis of how mid-training shapes post-training RL. We identify two key factors of mid-training algorithms that determine their effectiveness during RL: the efficiency of pruning the decision space and the effective planning horizon. The first factor governs the initial policy prior of RL, while the second decides the policy's potential to be improved through online interactions. To minimize regret in post-training RL, an ideal mid-training algorithm should extract from finite expert demonstrations the complete set of action subsets sufficient for all tasks, and enable fast selection among them during RL. Our results show that pruning efficiency is inversely related to the cardinality of the smallest near-optimal action subset, and that post-training RL converges faster when actions are temporally extended. These findings suggest that mid-training should operate in the space of action abstractions rather than primitive actions. Intuitively, learning

---

[*]Work done at Apple. [†]Equal advising.

high-level "skills" that are transferable across tasks yields a compact decision space and shortens the planning horizon, which makes pruning more efficient and RL more tractable.

To uncover this action hierarchy, we derive a temporal variational lower bound for the next-token prediction (NTP) objective. It can be optimized by iterative expectation-maximization, which involves a self-supervised RL step that uses the log-likelihood of expert data as reward to discover the hidden latent sequence, and a supervised fine-tuning step on the bootstrapped data. With an appropriate latent prior, the KL divergence enforces temporal consistency, ensuring that the latents function as coherent action abstractions. These design choices yield a scalable mid-training algorithm, Reasoning as Action Abstractions (RA3), where the KL penalty naturally determines when rational rollouts are necessary, thereby controlling computational cost.

We evaluate RA3 on code generation tasks using Qwen and Llama base models ranging in size from 1B to 8B. The mid-training dataset consists of 3.5M code snippets totaling 1B tokens. Our results show that fine-tuning on data bootstrapped with action abstractions substantially reduces cross-entropy loss and improves performance across multiple benchmarks, including HumanEval, MBPP, and their extended variants. On average, RA3 achieves a 4 point gain over NTP and an 8 point gain over the base models. Furthermore, RA3 accelerates RLVR convergence and attains higher asymptotic performance on HumanEval+, MBPP+, LiveCodeBench, and CodeForces. Together, these findings highlight the scalability and advantages of learning action abstractions in mid-training.

## 2 BACKGROUND

**Imitation Learning.** A task $\mathcal{M} = (\mathcal{S}, \mathcal{A}, R, \gamma)$ is an MDP defined by the state space $\mathcal{S}$, action space $\mathcal{A}$, reward $R$, and the discount factor $\gamma < 1$. In the language space, states are contexts that include all the previous tokens, and each action is a single token. The state transition is either deterministic by appending the new action tokens to the previous context or governed by the external environment. An expert policy is the policy that maximizes the expected state-action value:

$$\pi_E \in \operatorname*{argmax}_\pi \mathbb{E}_{a_t \sim \pi}\Big[R(s_t, a_t) + \gamma V_\mathcal{M}^\pi(s_{t+1})\Big] = \mathbb{E}_{a_t \sim \pi}\Big[\sum_t \gamma^t R(s_t, a_t)\Big].$$

It is worth noting that when the task is inherently solvable within one step, such as math problems where $r(s_0, a_{\mathrm{gt}}) = 1$, the expert policy should deterministically output the ground-truth answer $a_{\mathrm{gt}}$ at $s_0$, i.e., $\pi_E(a_{\mathrm{gt}}|s_0) = 1$, to maximize the return. Since most mid-training math data has explicit human reasoning before $a_{\mathrm{gt}}$, we instead focus on the multi-step decision-making domains, such as code generation and agentic tasks, where the expert trajectories are a sequence of actions.

Next-token prediction (NTP) during mid-training can be viewed as an imitation learning process on an offline expert dataset $\mathcal{D}_E$, collected by rolling out $\pi_E$ on the sampled tasks $\mathcal{M} \sim p(\mathcal{M})$. Its objective is to maximize the conditional log-likelihood:

$$\mathcal{J}_{\mathrm{NTP}}(\pi) = \mathbb{E}_{(s_{0:T}, a_{0:T}) \sim \mathcal{D}_E}\Big[\log \pi(a_{0:T} \mid s_{0:T})\Big] = \mathbb{E}_{\mathcal{D}_E}\left[\sum_{t=0}^T \log \pi(a_t \mid s_t)\right], \quad (2.1)$$

where $s_t \in \mathcal{S}$, $a_t \in \mathcal{A}$, $T$ is the total number of tokens in one expert demonstration, $\pi$ is the training policy, and $s_0$ is the beginning of the sentence (BOS) token. The formula in (2.1) applies directly to actions at coarser granularity than tokens, such as sentence-level actions.

NTP is adopted in different stages of LLM training, including pre-training, continued pre-training (or mid-training, if the goal is to acquire reasoning foundations before RLVR), and supervised fine-tuning. We primarily focus on the mid-training stage during the three-stage training procedure: pre-training, mid-training, and RLVR post-training.

**RL with Verifiable Reward.** The goal of post-training RL is to maximize the expected return. A common setup for RLVR is to use a binary outcome-based reward in a single-step MDP, defined as $r(s, o) = \mathrm{verifier}(s, o)$ to measure if the model response $o$ is identical to the ground-truth answer corresponding to the prompt question $s$. We adopt Group Relative Policy Optimization (GRPO) (Shao et al., 2024; Guo et al., 2025) as our default RLVR algorithm in experiments. Its objective is

$$\mathcal{J}_{\mathrm{GRPO}}(\pi) = \frac{1}{G}\sum_{i=1}^G \left(\min\left(\frac{\pi(o_i \mid s)}{\pi_{\mathrm{old}}(o_i \mid s)}A_i, \mathrm{clip}\left(\frac{\pi(o_i \mid s)}{\pi_{\mathrm{old}}(o_i \mid s)}, 1 \pm \epsilon\right)A_i\right) - \beta \mathcal{D}_{\mathrm{KL}}(\pi, \pi_{\mathrm{ref}})\right),$$

where $o_i \sim \pi_{\text{old}}(\cdot|s)$, $\epsilon, \beta$ are hyperparameters, and the advantage is calculated within the group $G$:

$$A_i = \left(r(s, o_i) - \text{mean}(\{r(s, o_i)\}_{i=1}^G)\right) / \left(\text{std}(\{r(s, o_i)\}_{i=1}^G)\right). \tag{2.2}$$

# 3 HOW MID-TRAINING SHAPES POST-TRAINING RL

The goal of post-training RL is to minimize the regret

$$\min_\pi \mathbb{E}_{\mathcal{M} \sim p(\mathcal{M})}[V_{\mathcal{M}}^*(s_0) - V_{\mathcal{M}}^\pi(s_0)], \tag{3.1}$$

i.e., to learn $\pi$ that is near-optimal across tasks $\mathcal{M}$.

An effective mid-training stage should extract from finite expert demonstrations the complete set of action subsets sufficient to solve all tasks, and facilitate efficient selection among them during RL for fast convergence. This perspective highlights that the efficiency of mid-training lies in pruning away useless actions and structuring the decision space so that online RL can improve policies effectively.

To formalize this, we define $\mathcal{M}_{\mathcal{Z}'} = (\mathcal{S}, \mathcal{Z}', R, \gamma)$ in a way similar to $\mathcal{M}$, except with action space restricted to $\mathcal{Z}'$. We say that $\mathcal{Z}'$ is near-optimal for $\mathcal{M}$ if near-optimal policies can be constructed using only the actions in $\mathcal{Z}'$. The goal of mid-training is thus to prune away with high probability all "bad" action subsets $\mathcal{Z}'$ that are sub-optimal. Formally, we define:

**Definition 3.1** (Near-Optimal Task Action Subset). An action subset $\mathcal{Z}' \subset \mathcal{Z}$ is called $\epsilon$-optimal for task $\mathcal{M}$ if the optimal values in $\mathcal{M}$ and $\mathcal{M}_{\mathcal{Z}'}$ satisfy $\Delta(\mathcal{M}, \mathcal{Z}') := V_{\mathcal{M}}^*(s_0) - V_{\mathcal{M}_{\mathcal{Z}'}}^*(s_0) \leq \epsilon$.

This definition allows us to connect mid-training with post-training RL through the following decomposition:

**Lemma 3.2** (Regret Decomposition). For any $\mathcal{Z}' \subseteq \mathcal{Z}$, the post-training RL regret in (3.1) satisfies

$$\mathbb{E}_{\mathcal{M} \sim p(\mathcal{M})}\left[V_{\mathcal{M}}^*(s_0) - V_{\mathcal{M}}^\pi(s_0)\right] = \underbrace{\mathbb{E}_{\mathcal{M} \sim p(\mathcal{M})}[\Delta(\mathcal{M}, \mathcal{Z}')]}_{\text{action-set pruning error}} + \underbrace{\mathbb{E}_{\mathcal{M} \sim p(\mathcal{M})}[V_{\mathcal{M}_{\mathcal{Z}'}}^*(s_0) - V_{\mathcal{M}_{\mathcal{Z}'}}^\pi(s_0)]}_{\text{post-training RL error}}.$$

Lemma 3.2 indicates that mid-training should identify an action subspace $\widehat{\mathcal{Z}}$ that minimizes two sources of error: (1) the approximation error from pruning the full action space $\mathcal{Z}$ down to $\widehat{\mathcal{Z}}$, and (2) the post-training RL error incurred when planning within $\widehat{\mathcal{Z}}$.

Corresponding to these two error terms, we demystify the key factors that determine the effectiveness of mid-training: (1) the efficiency with which the algorithm prunes the decision space from expert demonstrations, and (2) its impact on the convergence speed of post-training RL. The first factor governs the learned prior of the initial RL policy, while the second determines the extent to which the policy can be further improved through online interactions.

## 3.1 MID-TRAINING ACQUIRES STRONG POLICY PRIORS VIA EFFICIENT ACTION PRUNING

A central metric of mid-training pruning is how efficiently the algorithm can eliminate useless actions, since it directly determines the quality of the resulting action space given a fixed number of expert demonstrations. To formalize this, we introduce the following notion.

**Definition 3.3** (Minimal Size of Near-Optimal Action Subset). $\overline{\mathcal{Z}}_\epsilon \subset \mathcal{Z}$ is an $\epsilon$-optimal action subset if it is $\epsilon$-optimal for all tasks. Let $|\overline{\mathcal{Z}}_\epsilon|$ denote the minimal size of such a subset.

According to Definition 3.1, $\mathcal{A}$ itself is an optimal action set since $\Delta(\mathcal{M}, \mathcal{A}) = 0$. Thus, $\overline{\mathcal{Z}}_\epsilon$ always exists and $|\overline{\mathcal{Z}}_\epsilon|$ is finite. Now we are ready to give our first result on the pruning efficiency during mid-training.

**Theorem 3.4** (Pruning Efficiency). Denote $|\mathcal{D}_E|$ as the rollout number in the mid-training data. If

$$|\mathcal{D}_E| = \Theta\left(|\overline{\mathcal{Z}}_\epsilon| \log(|\mathcal{Z}|/\delta)/\sigma\right),$$

then with probability at least $1 - \delta$, the pruning error in Lemma 3.2 satisfies $\mathbb{E}_{\mathcal{M} \sim p(\mathcal{M})}[\Delta(\mathcal{M}, \widehat{\mathcal{Z}})] \leq \epsilon(1 - \sigma) + \sigma/(1 - \gamma)$.

We defer all proofs to Appendix A. Theorem 3.4 shows that the number of expert samples required for action-space pruning during mid-training decreases as both $|\overline{\mathcal{Z}}_\epsilon|$ and $|\mathcal{Z}|$ shrink. With a higher pruning efficiency, i.e., a smaller $|\overline{\mathcal{Z}}_\epsilon|$ and $|\mathcal{Z}|$, the probability that suboptimal actions survive mid-training decreases, leading to a smaller pruning error. This result highlights the importance of a compact decision space and motivates defining $\mathcal{Z}$ as a space of action abstractions instead of primitive actions $\mathcal{A}$.

Action abstractions are defined analogously to Markov options (Puterman, 1994; Sutton et al., 1999; Precup, 2000), representing the abstraction of temporally-extended primitive actions. Each $z \in \mathcal{Z}$ corresponds to a high-level intention that executes a sequence $a_i, \ldots, a_{i+\tau}$, with duration $\tau \sim p(\cdot \mid s, z)$. Notably, the above result for $\mathcal{Z}$ also applies to $\mathcal{A}$ as a special instantiation of $\mathcal{Z}$, with $\tau = 1$ and $a_j = z_j$.

Compared to the space of primitive actions $\mathcal{A}$, the space of action abstraction has a substantially smaller size of $|\mathcal{Z}|$ and $|\overline{\mathcal{Z}}_\epsilon|$, since each action abstraction corresponds to a transferable skill spanning multiple tasks. Consequently, learning action abstractions enables mid-training to more efficiently approximate a near-optimal action set and provide a stronger policy prior, thereby reducing the burden on post-training RL.

## 3.2 Mid-Training Accelerates RL Convergence with Shorter Decision Horizon

The above analysis mainly considers the influence of mid-training on the pruning error and fixed the post-training RL error. Next, we analyze how mid-training shapes post-training RL. Our result below is based on value iteration due to its simplicity.

**Theorem 3.5** (RL Convergence Rate). To achieve an $\varepsilon$-optimality that satisfies $\|V_N - V^*\|_\infty \leq \varepsilon$, the required number of iterations $N$ is lower-bounded by $N \geq \frac{1}{1-\overline{\gamma}} \log \frac{R_{\max}}{\varepsilon(1-\gamma)}$, where $\overline{\gamma} = \sup_{s,0} \mathbb{E}[\gamma^\tau | s, z] \leq \gamma$ and $R_{\max} = \max_{s,a} R(s, a)$.

The above result reveals that the reasoning structures acquired during mid-training influence the convergence through the duration $\tau$ of the temporally-extended actions. For actions that last longer, $\overline{\gamma}$ is smaller and RL converges faster to optimality than mid-training with NTP, where $\tau = 1$ and $\overline{\gamma} = \gamma$. This makes intuitive sense as each Bellman backup jumps across $\tau$ steps in one shot, which shortens the effective planning horizon and shrinks the error faster per iteration. Similar $1/(1 - \overline{\gamma})$ dependency also appears in the bound for broader RL algorithms such as policy gradient (Agarwal et al., 2021; Zhang et al., 2023), which we omit due to the requirements of additional assumptions.

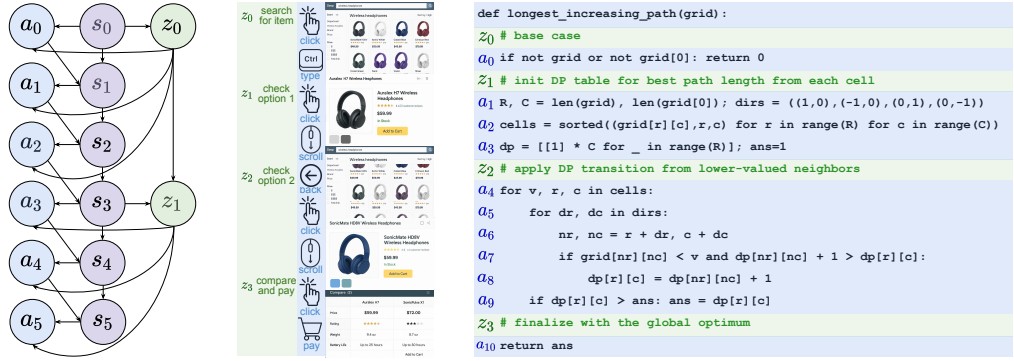

Figure 1: **(Left)**: The probabilistic graphical model of the action hierarchy. **(Middle & Right)**: Examples of primitive actions in expert demonstrations (blue) and the hidden high-level temporal abstractions (green), in web agent and code generation domains, respectively.

## 4 From Primitive Actions to Temporal Abstractions

In Section 3, we analyze the benefits of leveraging temporal action abstractions during mid-training from two perspectives: its efficiency in pruning the action space and its ability to accelerate subsequent RL. Intuitively, there are high-level "skills" that are shared across tasks, utilizing which helps

generalization and makes planning easier, since the action space shrinks and the decision horizon reduces. We illustrate this with two examples in Figure 1. In what follows, we introduce a principled way to extract the temporal abstractions from the primitive actions in the mid-training data.

## 4.1 TEMPORAL VARIATIONAL BOUND

We begin by seeking an alternative way to maximize the likelihood beyond predicting the next tokens. Specifically, we give a sequential Evidence Lower Bound (ELBO) of the NTP objective:

**Theorem 4.1** (Temporal ELBO). The next-token prediction objective in (2.1) is lower bounded by

$$\mathcal{J}_{\text{NTP}}(\pi) \geq \mathcal{J}(\pi, q) = \mathbb{E}_{(s_{0:T}, a_{0:T}) \sim \mathcal{D}_E, z_t \sim q} \left[ \sum_{t=0}^{T} \log \pi(a_t | s_t, z_t) - \mathcal{D}_{\text{KL}} \Big( q(z_t | s_t, z_{0:t-1}) \, \| \, p(z_t | s_t, z_{0:t-1}) \Big) \right],$$

where $p(z_t | s_t, z_{t-1})$ is the prior distribution of $z_t$.

The ELBO introduces a sequence of latents $z_{0:T}$ to model the observed primitive actions $a_{0:T}$. The intuition behind it is that there are hidden thoughts or intentions, i.e., $z_{0:T}$, behind the expert decisions $a_{0:T} \sim \mathcal{D}_E$, which are not present in the mid-training data. The amortized variational inference leverages the parameterized $q(z_t | s_t, z_{0:t-1})$ as an approximation of the true posterior of $z$. Optimizing the ELBO is equivalent to modeling the distribution of the latents $z_{0:T}$ and using the inferred latents to better model the likelihood $p(a_{0:T} | s_{0:T})$.

We maximize the ELBO by alternatively optimizing $q$ and $\pi$, in an Expectation-Maximization (EM) manner. In each EM iteration $i$, the E step fixes $\pi_i$ and updates $q$ to maximize the ELBO, i.e.,

$$q_i = \underset{q}{\arg\max} \, \mathcal{J}(\pi_i, q) = \underset{q}{\arg\max} \, \mathbb{E}_{\mathcal{D}_E, z_t \sim q} \left[ \sum_{t=0}^{T} \underbrace{\log \pi_i(a_t | s_t, z_t) - \mathcal{D}_{\text{KL}} \big( q \, \| \, p \big)}_{\text{RL reward at step } t} \right]. \tag{4.1}$$

This corresponds to a $T$-horizon RL procedure with the per-step reward defined as the log-likelihood of the observed expert actions, with a KL penalty that we will discuss later. Intuitively, (4.1) encourages the sequence of latents $z_{0:T}$ sampled from $q_i$ to "explain" the expert decisions $a_{0:T}$.

The $M$-step then fixes the updated $q_i$ and optimizes $\pi$:

$$\pi_{i+1} = \underset{\pi}{\arg\max} \, \mathcal{J}(\pi, q_i) = \underset{\pi}{\arg\max} \, \mathbb{E}_{\mathcal{D}_E, z_t \sim q_i} \big[ \log \pi(a_t | s_t, z_t) \big], \tag{4.2}$$

which is simply imitating the expert trajectories that are bootstrapped with the inferred latents $z_t$.

## 4.2 TEMPORALLY-CONSISTENT LATENTS AS ACTION ABSTRACTIONS

We have transformed the maximization of likelihood on primitive actions $a_{0:T}$ to an ELBO objective that learns from the bootstrapped sequence with latent trajectories sampled from the variational posterior $q(z_t | s_t, z_{0:t-1})$. Recall that our analyses in Section 3 reveal the benefits of learning compact sets of high-level action abstractions. We will show in the following how to fulfill this goal with a properly defined latent prior.

Specifically, the latent $z_t$ in the ELBO is defined per-step for every $t \in [0, T]$. To let $z_t$ represent an abstraction of temporally-extended actions that spans across $\tau \sim p(\cdot | s_t, z_t)$ timesteps, it is equivalent to let $z_t = z_{t+1} = \cdots = z_{t+\tau}$[*]. This can be achieved by setting the prior $p(z_{t+1} | s_{t+1}, z_t)$ in Theorem 4.1 to have a large probability mass on $z_t$, and uniformly distributed at all other positions:

$$p(z_t | s_t, z_{t-1}) = \alpha \delta(z_{t-1}) + (1 - \alpha) U(z_t), \tag{4.3}$$

where $\alpha \in [0, 1]$ is a hyperparameter, $\delta(\cdot)$ is the Dirac delta function, and $U(\cdot)$ is the uniform distribution over $\mathcal{Z}$. The delta function helps preserve a temporally-consistent latent as a high-level action abstraction, and the uniform distribution encourages learning a diverse reasoning foundation.

---

[*]The notation here deviates slightly from Section 2, where all $\tau$ identical latents are written as a single $z$.

## 5 RA3: A SCALABLE MID-TRAINING ALGORITHM

In the context of LLMs, the action abstraction $z$ serves a similar role as the rational or intention, and the primitive action $a$ is the actual answer or operation in the environment. Taking code generation as an example, $z_t$ can be the reasoning before writing the next code block $a_{t+\tau}$.

In addition to the theoretical benefits discussed in Section 3, the temporal consistency of the latents also makes it possible to scale up the optimization on mid-training-sized data. Specifically, generating rationals $z_{0:T}$ corresponds to $T$ rollouts, whose size is proportional to the total number of tokens in the data. This prohibits us from making full use of the high-quality data in the mid-training corpus, which typically contains billions of tokens. Fortunately, the temporal consistency of latents rescues us from sampling rollout $z_t$ at each timestep $t$, as it is kept unchanged for $\tau$ steps.

To avoid redundant sampling, we define two types of latents: $z = \text{<act>}$ and $z$ that begins with <think>. For latents $z_t = z_{t-1}$ that are temporally-consistent, $a_t$ is directly generated without sampling new rationals since it fall under the same high-level intention as $a_{t-1}$. For this reason, we use $z_t = \text{<act>}$ to indicate $z_t = z_{t-1}$. Whenever <act> is sampled, the rollout stops immediately. By doing so, full rollouts are only sampled when <think> is the first token, significantly reducing the RL inference cost. Besides, we may rewrite the KL term in the ELBO as follows.

**Proposition 5.1.** With the prior defined in (4.3), the KL term in Theorem 4.1 satisfies

$$\mathcal{D}_{\text{KL}}\Big(q(z_t|s_t, z_{0:t-1}) \,||\, p(z_t|s_t, z_{0:t-1})\Big) = \mathcal{D}_{\text{KL}}\Big(\text{Bern}(q_{\text{act}}) \,||\, \text{Bern}(\alpha)\Big) - (1 - q_{\text{act}})\mathcal{H}\Big(q(z_t|s_t, z_{0:t-1})\Big),$$

where $q_{\text{act}} = q(z_t = \text{<act>} \mid s_t, z_{0:t-1})$, $\text{Bern}(\cdot)$ is the Bernoulli distribution, and

$$\mathcal{D}_{\text{KL}}\Big(\text{Bern}(q_{\text{act}}) \,||\, \text{Bern}(\alpha)\Big) = \mathbb{E}_{z \sim q}\Big[\mathbb{1}(z_t = \text{<act>}) \log \frac{q_{\text{act}}}{\alpha} + \mathbb{1}(z_t \neq \text{<act>}) \log \frac{1 - q_{\text{act}}}{1 - \alpha}\Big].$$

The KL is decomposed into two terms: a KL term between Bernoulli distributions and an entropy term that encourages diversity. By setting $\alpha > q_{\text{act}}$, the KL discourages unnecessary thinking since it assigns a larger penalty to $z_t \neq \text{<act>}$ than $z_t = \text{<act>}$. The penalty difference defines a threshold, guiding the model to generate new rationals only when they improve the log-likelihood reward by more than this threshold. In implementation, instead of tuning $\alpha$, we apply reward shaping and set a fixed penalty $c$ only to $z_t \neq \text{<act>}$. The proposition also indicates that the additional training cost compared to NTP can be adjusted by $\alpha$. In the extreme case where $\alpha = 1$, our algorithm degenerates to NTP, since $\mathcal{D}_{\text{KL}}(\text{Bern}(q_{\text{act}}) \,||\, \text{Bern}(1))$ is infinite for all $q_{\text{act}} < 1$, i.e., the $q$ policy receives an infinite penalty for generating any rationals. We provide the pseudocode in Algorithm 1.

---

**Algorithm 1** Reasoning as Action Abstractions (RA3) for Mid-Training

1: **Input:** Base LLM $\pi_0$, mid-training dataset $\mathcal{D}_E$, penalty hyperparameter $c$.
2: **for** EM iteration $i$ in $1, 2, \cdots$ **do**
3:      Optimize $q_i = \arg\max_q \mathbb{E}_{\mathcal{D}_E^{e_i}, z_t \sim q}[\sum_{t=0}^{T} \log \pi_i(a_t|s_t, z_t) - c\mathbb{1}(z_t \neq \text{<act>})]$ via RL.
4:      Fine-tune $\pi_{i+1} = \arg\max_\pi \mathbb{E}_{\mathcal{D}_E^{m_i}, z_t \sim q_i}\big[\log \pi(a_t \mid s_t, z_t)\big]$ via NTP.

---

We enforce the two types of actions discussed above by incorporating a simple format reward in the RL step that assigns zero rewards for wrong formats, which we omit in the pseudocode for clarity. We optimize the $T$-horizon RL using policy gradient, with advantages calculated in a similar way to (2.2) in GRPO: after sampling $G$ length-$T$ rollouts, we set the baselines as $b(s_{t'}) = \sum_{g=1}^{G} \sum_{t=t'}^{T} r_t^g / G$ that are independent on the actions, and combine it with the state-action value to calculate the advantage at each step.

## 6 RELATED WORK

**LLM Mid-Training.** RL has long been utilized for training language models (Nguyen et al., 2017; Paulus et al., 2017; Jaques et al., 2020; Ramamurthy et al., 2022; Ouyang et al., 2022). However, its potential as a universal policy-improvement operator has not been fully unlocked until recently, when reasoning models learn to cast intermediate thoughts as actions and optimize them via RL (Guo et al., 2025; Zeng et al., 2025b; Liu et al., 2025; Team et al., 2025). This paradigm also guides

the design of the policy prior. To obtain strong initial policies in terms of both performance and exploration diversity, mid-training (Xu et al., 2025; Wang et al., 2025c; Su et al., 2025) plays an important role, which performs reasoning or agentic continued pre-training on high-quality expert data. Previous work that leverages next-token prediction during mid-training is an imitation learning process, where the data comes from rollouts of optimal expert policies, such as humans' demonstrations of device-controlling and code-writing (Rawles et al., 2023; Huang et al., 2024; Bai et al., 2024). Anchoring our findings, it is observed that learning the action hierarchy with abstraction-based reasoning performs better than training on primitive actions alone (Xu et al., 2024; Chen et al., 2024; Wang et al., 2025b; Xue et al., 2025). For this reason, some mid-training datasets augment the expert data with synthetic reasoning distilled from frontier LLMs (Wang et al., 2025b; LI et al., 2024). However, the distributional shift makes it unclear how well the student LLM can benefit from these action abstractions, compared to RA3, which learns its own reasoning via RL. Besides, RA3 is more preferred considering the cost of augmenting trajectories with reasoning distillation on a large scale. In fact, for the code generation domain that we are interested in, most mid-training-sized datasets mainly contain human code from the internet, without costly relabeling.

**Self-Supervised RL.** Optimizing the temporal variational bound involves a self-supervised RL step with log-prob of the expert action as reward (Zhong et al., 2025; Ruan et al., 2025; Zhou et al., 2025a; Dong et al., 2025; Wang et al., 2025a). Compared to BRiTE (Zhong et al., 2025) that is formulated as a post-training method with a standard ELBO for single-step problems, we are motivated to learn action abstractions with the ELBO derived for temporal sequences, guided by our theoretical analysis. When the decision horizon is one, RA3 reduces to BRiTE. However, for multi-step problems such as code generation, naïvely applying BRiTE corresponds to treating the log-probability of the entire code sequence as a single RL reward, which leads to high variance and unstable training in our setup. Moreover, Dong et al. (2025) hand-crafts an entropy-based rule to determine reasoning positions and trains on only 4k samples with instructed reasoning models. In contrast, RA3 is theoretically grounded, scales to mid-training setups, and enables the model to autonomously learn when to skip unnecessary reasoning by generating temporally consistent latents.

**Markov Options.** RL based on options (Sutton et al., 1999; Precup, 2000; Bacon et al., 2017; Zhang et al., 2021) enables agents to represent courses of actions at extended time scales and learn in the MDP with them. It greatly helps long-horizon complex tasks by naturally introducing hierarchical structures (Jong et al., 2008). Our analysis of the decision space size is partly inspired by (Brunskill & Li, 2013; 2014). Different from their study on option transfer in lifelong and multi-task RL, we are mainly interested in mid-training algorithm design and its impact on post-training RL in LLMs.

## 7 EXPERIMENTS

**Experiment Setups.** We focus on Python code generation in our experiments. The granularity of primitive actions $a$ is a single line of code. For the two types of latents $z$, to avoid additional fine-tuning on special tokens, we remove the newline character \n at the end of $a$ and set <act>=\n, <think>=\n#. That is, after line $a_t$, the model either *only* outputs \n before $a_{t+1}$ or generates a comment line as a high-level abstraction to guide the code writing. The format reward is non-zero for the think action if it begins and ends with \n and the first non-space token is #. This design ensures that the reasoning bootstrapped data has the correct syntax. The RL step of RA3 is implemented in a similar way to multi-turn RL: the code line $a_t$ from the training corpus and the reasoning comment $z_t$ alternate as turns. We adopt asynchronous rollout with the SGLang engine (Zheng et al., 2024), as batched inference often incurs idle time between turns. That is, batched rollouts must wait for all $z_t$ to be generated before proceeding to the next turn, which is especially undesirable in our setting, where temporal consistency in $z$ is intended to reduce unnecessary reasoning.

We implement RA3 on three pre-trained models: Qwen-2.5-1.5B (Hui et al., 2024), Llama-3.2-1B, and Llama-3.1-8B (Dubey et al., 2024). For the mid-training data, we select the Python data within the continued pre-training corpus of Huang et al. (2024), which consists of a large portion of high-quality internet data (2.36M code snippets, 834M tokens) and a small portion of code-only synthetic data (129K snippets, 120M tokens). This leads to a total mid-training size of 3.5M code snippets with 1B tokens. Each EM iteration involves 400 gradient updates, with the first 40 being the RL policy gradient. The hyperparameters of our E step is the same as NTP: batch size of 512 with a learning rate of $2e-5$. The RL step of RA3 sets the maximum length of $z$ as 16, with a sampling

temperature of $1.0$ and a group size of $G = 3$. The entropy coefficient is set to $0.001$ and we do not regularize the reference KL. The RL batch size is set to $1024$ with a learning rate of $2e-6$. We set the penalty $c = 0.05$. AdamW optimizer is used (Loshchilov & Hutter, 2017) in experiments.

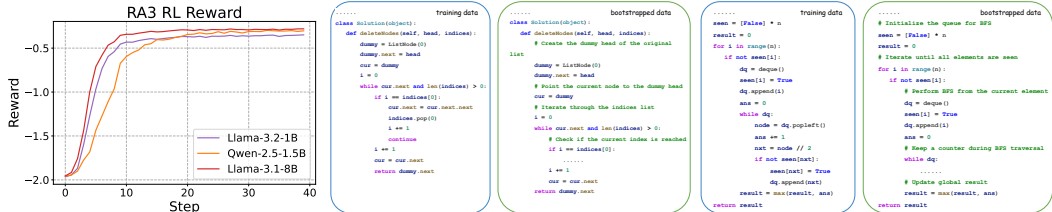

Figure 2: **(Left)**: Curve of RL training reward. **(Right)**: Examples of the data from mid-training and after reasoning bootstrapping, where skills are abstracted, such as dummy head creation and BFS.

**Mid-Training Results.** We begin by analyzing the results during the EM steps of RA3. The RL training reward in the first E step is given in Figure 2. We find that the model can quickly learn to maximize the reward, so that most of the compute can be allocated to reasoning bootstrapping and fine-tuning in the M step. We also provide two examples of data in Figure 2.

For the M step, an important metric is the cross-entropy (CE) loss, i.e., the negative log-likelihood of the next token. It can be observed from Figure 3 that fine-tuning on reasoning-bootstrapped data significantly accelerates learning speed. This supports the scope of mid-training we defined in Section 2: the expert policy rollouts are sequences of primitive actions, such as the raw coding lines. There are hidden reasoning trajectories that the expert follows to make decisions, but are unavailable in the data. The learning is much easier when such reasoning traces are extracted via RL.

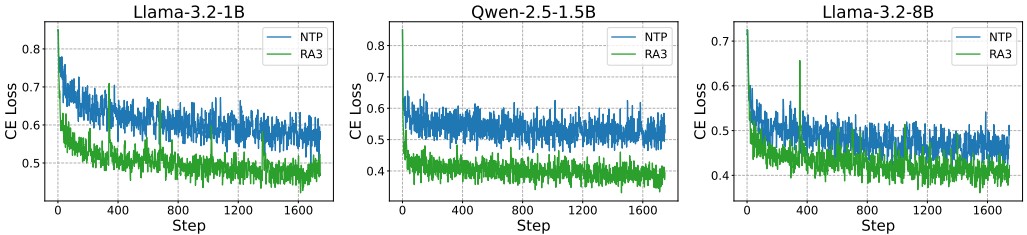

Figure 3: Data bootstrapped with reasoning learned in the E step reduces the CE loss in the M step.

To better assess the algorithms, we evaluate the resulting models on the HumanEval (Chen et al., 2021), MBPP (Austin et al., 2021), HumanEval+, and MBPP+ (Liu et al., 2023) benchmarks using Ben Allal et al. (2022). We test the improvement of each EM iteration of RA3 by the average scores on these benchmarks, and compare with the NTP checkpoints that are trained on the same data. The results are shown in Figure 4. It can be observed that learning to reason as action abstractions not only gives lower CE loss, but also achieves higher evaluation performance with fewer data.

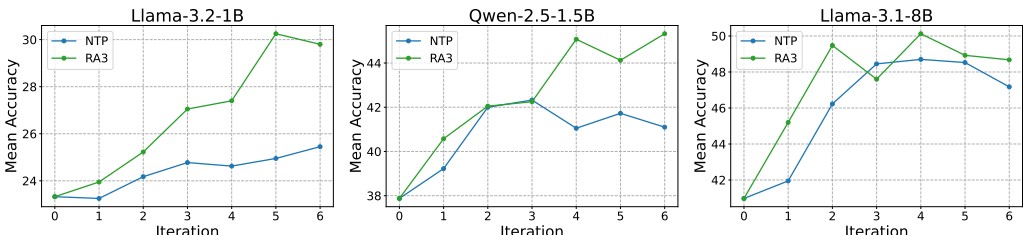

Figure 4: Evaluation results during mid-training, with accuracies averaged across four benchmarks.

We also report the pass@k results of the best checkpoint for NTP and RA3 in Table 1. The result reveals the generalization benefits of learning with temporal action abstractions.

Table 1: Mid-training performance: pass@k on HumanEval, MBPP, HumanEval+, and MBPP+.

| Model | Method | HumanEval | | MBPP | | HumanEval+ | | MBPP+ | | Avg. | |
|---|---|---|---|---|---|---|---|---|---|---|---|
| | | p@1 | p@5 | p@1 | p@5 | p@1 | p@5 | p@1 | p@5 | p@1 | p@5 |
| Llama 3.2 1B | Base | 18.9 | 30.4 | 25.8 | 37.0 | 17.1 | 25.7 | 31.5 | 47.9 | 23.3 | 35.3 |
| | NTP | 21.3 | 34.1 | 27.8 | 45.8 | 17.7 | 29.5 | 34.4 | 51.8 | 25.3 | 40.3 |
| | RA3 | **25.0** | **38.1** | **32.8** | **46.1** | **22.0** | **33.5** | **39.4** | **54.6** | **29.8** | **43.1** |
| Qwen 2.5 1.5B | Base | 37.2 | 53.7 | 38.6 | 56.5 | 32.3 | 48.1 | 43.4 | 61.0 | 37.9 | 54.8 |
| | NTP | 41.5 | 57.3 | 43.4 | 59.2 | 35.4 | 53.3 | 46.6 | 64.8 | 41.7 | 58.7 |
| | RA3 | **48.2** | **63.6** | **45.8** | **61.3** | **42.7** | **59.0** | **49.7** | **64.9** | **46.6** | **62.2** |
| Llama 3.1 8B | Base | 36.6 | 57.5 | 45.2 | 59.1 | 30.5 | 54.7 | 51.6 | 65.7 | 41.0 | 59.3 |
| | NTP | 48.2 | 62.0 | **48.6** | 62.9 | 42.7 | 60.1 | 51.1 | 67.4 | 47.7 | 63.1 |
| | RA3 | **50.0** | **66.2** | 48.0 | **63.0** | **44.5** | **61.3** | **53.2** | **67.8** | **48.9** | **64.6** |

**Post-Training RLVR Results.** We then study the impact of mid-training on post-training RL. We use GRPO as the default RLVR algorithm, and leverage the off-the-shelf DeepCoder codebase (Luo et al., 2025). We use the AReaL-boba-2-RL-Code dataset (Fu et al., 2025) for training, which contains 7.7K data, filtered from TACO (Li et al., 2023), Mattern et al. (2025), and LiveCodeBench (Jain et al., 2024). We run independent RLVR training with different random seeds (3 for small models and 2 for the 8B model), and the evaluation results are reported in Figure 5.

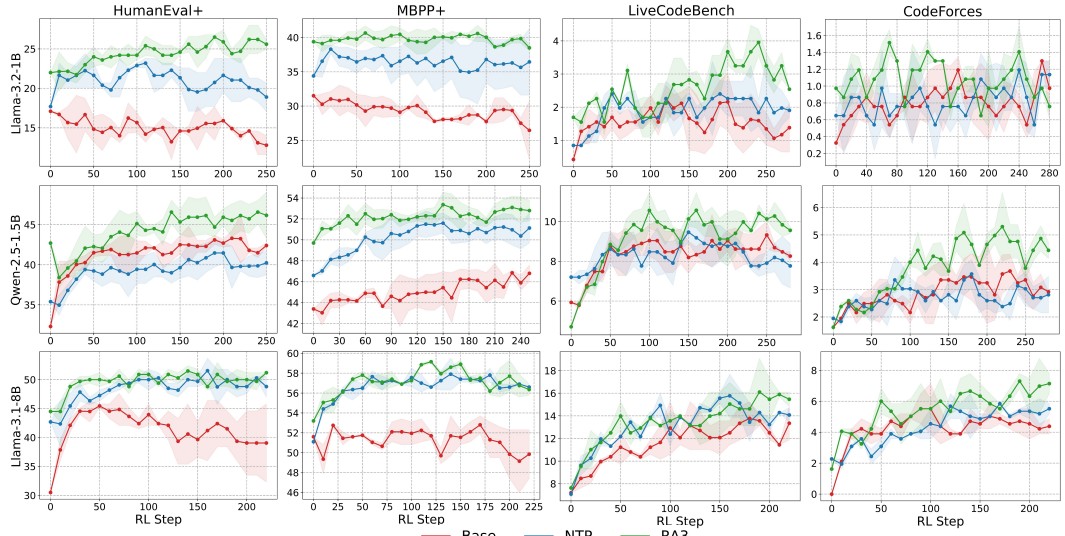

Figure 5: RLVR training results (mean and standard error) from different mid-training models. We observe that mid-training significantly improves the RLVR performance compared to the base models. Besides, RA3 is able to learn more effectively during RLVR, in terms of both convergence speed and asymptotic performance, supporting our theoretical results in Section 2.

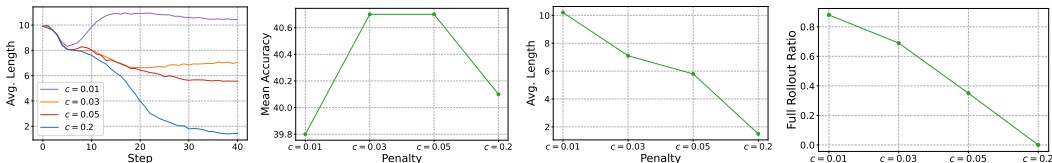

Figure 6: We study the effect of penalty $c$ on (a) the behavior of the RL step, (b) the mean accuracy of HumanEval and MBPP, (c) the average length of $z$, and (d) the ratio of full rollout samples.

**Ablation Study.** We first investigate the effect of the KL penalty $c$ when training the Qwen model. Recall that $c$ regulates the temporal consistency of the latent action $z$ by setting a threshold for when

additional reasoning is necessary. As shown in Figure 6, a small $c$ causes the $q$-policy to generate redundant $z$ values at most timesteps. This behavior is expected: for almost every step, there exists some $z_t \neq$ <act> that explains $a_{t+1}$ more effectively (i.e., yields higher log-likelihood RL reward) than the null action $z_t =$ <act>. However, such behavior offers no advantage over primitive-action NTP, since neither the effective planning horizon nor the decision space is reduced. Moreover, the computational overhead of RA3 can be controlled through $c$: a large penalty $c = 0.2$ requires that <think> improves the expert log-likelihood by at least $0.2$ to be rewarded, an unlikely event. In this regime, the policy learns to output only <act>, and RA3 has nearly identical computational cost as NTP. In our default setup $c = 0.05$, RA3 is efficient by generating less than $40\%$ full rollouts with an average length of less than $6$, adding only modest overhead compared to NTP while still providing substantial performance gains. This offers a practical knob to balance performance and compute when scaling RA3.

Besides, we construct an additional baseline that performs NTP on synthetic rationals. We first fine-tune a separate LLM to generate high-quality reasoning trajectories in the form of single-line comments. It is trained using 366K synthetic Python code snippets from Huang et al. (2024), which are collected by prompting the expert LLM to generate self-contained code along with reasoning comments. Then for the Qwen-2.5-1.5B model, we perform NTP on training data bootstrapped by this reasoning-synthesis model. It can be observed that NTP on synthetic rationals underperforms our method. We attribute this to a key difference in how the reasoning traces are obtained: in RA3, the latents are learned

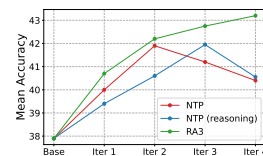

Figure 7: Comparison with synthetic reasoning NTP baselines.

by the model itself via an RL-regularized ELBO objective and are proven to form a lower bound on the conditional log-likelihood. As a result, the learned abstractions are more amenable to further optimization and also have better learnability than external traces.

## 8 CONCLUSION

An effective mid-training stage should extract from expert demonstrations the action subsets sufficient for all tasks and structure the decision space to enable fast, efficient reinforcement learning (RL) convergence. In this paper, we present the first formal analysis of how mid-training design choices influence post-training RL. Our findings highlight two critical roles mid-training should play: (i) mid-training should perform efficient action space pruning, which determines the learned policy prior, and (ii) mid-training should accelerate subsequent RL convergence with shorter effective planning horizons, which reflects the policy's potential for improvement through online interaction. We show that both perspectives favor learning in the space of temporally-extended actions over the large space of primitive actions used in next-token prediction. Building on these insights, we propose a novel mid-training algorithm based on the temporal variational bound. This method iteratively uncovers temporally consistent reasoning trajectories via RL and fine-tunes on the resulting bootstrapped data. Experiments on code generation tasks validate the advantages of reasoning as action abstraction, improving both the generalizability of the initial policy and the effectiveness of subsequent RLVR, in terms of convergence speed and asymptotic performance.

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

## A PROOFS

### A.1 PROOF OF LEMMA 3.2

*Proof.* Since the policy $\pi$ selects actions from $\widehat{\mathcal{Z}}$, it is an admissible policy in both $\mathcal{M}$ and $\mathcal{M}_{\widehat{\mathcal{Z}}}$. Since the dynamics, rewards, and discount factors are identical for $\mathcal{M}$ and $\mathcal{M}_{\widehat{\mathcal{Z}}}$, we have $V_{\mathcal{M}}^{\pi} = V_{\mathcal{M}_{\widehat{\mathcal{Z}}}}^{\pi}$. The results then follow from basic algebra. $\square$

### A.2 PROOF OF THEOREM 3.4

*Proof.* Before the proof, we first define the subset of suboptimal actions.

**Definition A.1** (Suboptimal Action Subset). $\mathcal{Z}'$ is $(\epsilon, \sigma)$-suboptimal if $\mathbb{E}_{\mathcal{M} \sim p(\mathcal{M})}[\mathbb{1}(\Delta(\mathcal{M}, \mathcal{Z}') > \epsilon)] \geq \sigma$.

For an $(\epsilon, \sigma)$-bad action subset $\mathcal{Z}'$, according to Definition A.1, we know that

$$\Pr_{\mathcal{M} \sim p(\mathcal{M})}[\Delta(\mathcal{M}, \mathcal{Z}') \geq \epsilon] \geq \sigma.$$

Therefore, for a randomly drawn task $\mathcal{M} \sim p(\mathcal{M})$, the probability that $\mathcal{Z}'$ is not pruned away is no more than $1 - \sigma$.

With $|\mathcal{D}_E|$ independent expert demonstrations in mid-training, we reject $\mathcal{Z}'$ if $V_{\mathcal{M}^i}^*(s_0) < V_{\mathcal{M}_{\mathcal{Z}'}^i}^*(s_0) + \epsilon$ for any $i \in [1, |\mathcal{D}_E|]$. The probability that $\mathcal{Z}'$ is not pruned away during mid-training is thus no more than $(1 - \sigma)^{|\mathcal{D}_E|}$. We denote it as

$$p(\text{survive}(\mathcal{Z}')) \leq (1 - \sigma)^{|\mathcal{D}_E|} \leq e^{-\sigma|\mathcal{D}_E|},$$

where the second inequality holds since $1 + x \leq e^x$ for all real $x$.

Let $\mathcal{B} = \{\widehat{\mathcal{Z}} \subseteq \mathcal{Z} : |\widehat{\mathcal{Z}}| = |\overline{\mathcal{Z}}|\}$ be the set of all $|\overline{\mathcal{Z}}|$-length subsets. We care about the event that there exist some bad subsets of actions that are not pruned away, i.e., survive:

$$p\left(\bigcup_{\mathcal{Z}' \in \mathcal{B}} \text{survive}(\mathcal{Z}')\right) \leq \sum_{\mathcal{Z}' \in \mathcal{B}} p(\text{survive}(\mathcal{Z}')) \leq \binom{|\mathcal{Z}|}{|\overline{\mathcal{Z}}|} e^{-\sigma|\mathcal{D}_E|} = \Theta\left(|\mathcal{Z}|^{|\overline{\mathcal{Z}}|}\right) e^{-\sigma|\mathcal{D}_E|},$$

where the inequalities hold by applying a union bound over all candidate subsets.

Plugging in $|\mathcal{D}_E| = \Theta(|\overline{\mathcal{Z}}| \log(|\mathcal{Z}|/\delta)/\sigma)$ gives us

$$p\left(\bigcup_{\mathcal{Z}' \in \mathcal{B}} \text{survive}(\mathcal{Z}')\right) \leq \delta,$$

i.e., all the $(\epsilon, \sigma)$-bad action subsets $\mathcal{Z}'$ are pruned away from $\mathcal{Z}$ with probability at least $1 - \delta$.

In other words, any action subset $\widehat{\mathcal{Z}}$ after pruning is *not* $(\epsilon, \sigma)$-suboptimal with high probability, i.e., it satisfies $\mathbb{E}_{\mathcal{M} \sim p(\mathcal{M})}[\mathbb{1}(\Delta(\mathcal{M}, \widehat{\mathcal{Z}}) > \epsilon)] < \sigma$, or equivalently $\mathbb{E}_{\mathcal{M} \sim p(\mathcal{M})}[\mathbb{1}(\Delta(\mathcal{M}, \widehat{\mathcal{Z}}) \leq \epsilon)] \geq 1 - \sigma$, with probability at least $1 - \delta$.

Since the rewards are in $[0, 1]$, we know that $V_{\mathcal{M}}^*(s_0) \leq \sum_{t=0}^{\infty} \gamma^t = 1/(1 - \gamma)$. Thus, $\Delta(\mathcal{M}, \widehat{\mathcal{Z}}) \leq 1/(1 - \gamma)$.

Therefore, with probability at least $1 - \delta$, the pruning error $\mathbb{E}_{\mathcal{M} \sim p(\mathcal{M})}[\Delta(\mathcal{M}, \widehat{\mathcal{Z}})] \leq \epsilon(1 - \sigma) + \sigma/(1 - \gamma)$. $\square$

### A.3 PROOF OF THEOREM 3.5

*Proof.* We first define the value iteration operator $\mathcal{T}$ in the temporally extended action space $\mathcal{Z}$ as

$$(\mathcal{T}V)(s) = \max_{z \in \mathcal{Z}} \left\{ \mathbb{E}_{\tau, s'}[R_\tau + \gamma^\tau V(s') \mid s, z] \right\},$$

where $p(s'|s, z) = \sum_j \gamma^j p(s', \tau = j|s, z)$, $p(s', \tau = j|s, z)$ is the joint probability of transitioning to $s'$ in $\tau$ steps after taking action $z$ at $s$, and $R_\tau = \sum_{k=1}^\tau \gamma^k R_k$.

For any functions $V_1$, $V_2$ and any state $s$, we have

$$(\mathcal{T}V_1)(s) - (\mathcal{T}V_2)(s) = \max_{z \in \mathcal{Z}}\Big\{\mathbb{E}_{\tau,s'}\big[R_\tau + \gamma^\tau V_1(s') \mid s, z\big]\Big\} - \max_{z \in \mathcal{Z}}\Big\{\mathbb{E}_{\tau,s'}\big[R_\tau + \gamma^\tau V_2(s') \mid s, z\big]\Big\}$$
$$\leq \max_{z \in \mathcal{Z}}\mathbb{E}\big[\gamma^\tau V_1(s') - V_2(s') \mid s, z\big],$$

where the inequality holds since $\max f - \max g \leq \max(f - g)$ for any $f, g$.

Let $\|\cdot\|_\infty$ be the sup norm on functions $V : \mathcal{S} \mapsto \mathbb{R}$. Then

$$(\mathcal{T}V_1)(s) - (\mathcal{T}V_2)(s) \leq \max_{z \in \mathcal{Z}}\Big\{\mathbb{E}\big[\gamma^\tau\big] \cdot \|V_1 - V_2\|_\infty\Big\} \leq \overline{\gamma}\,\|V_1 - V_2\|_\infty .$$

Since the above inequality holds for all $s \in \mathcal{S}$, taking the supremum over $s$ gives us $\|\mathcal{T}V_1 - \mathcal{T}V_2\|_\infty \leq \overline{\gamma}\,\|V_1 - V_2\|_\infty$. Therefore, $\mathcal{T}$ is a $\overline{\gamma}$-contraction on $(\mathbb{R}^\mathcal{S}, \|\cdot\|_\infty)$.

By Banach's fixed-point theorem, $\mathcal{T}$ has a unique fixed point $V^*$ and value iteration $V_{n+1} = \mathcal{T}V_n$ converges geometrically:

$$\|V_n - V^*\|_\infty \leq \overline{\gamma}^n \|V_0 - V^*\|_\infty \leq \overline{\gamma}^n R_{\max}/(1 - \gamma),$$

where the last inequality holds since $V_0 = 0$ and the maximum value satisfies $V_{\max} = \sum_t \gamma^t R_{\max} = R_{\max}/(1 - \gamma)$.

In order to get $\|V_N - V^*\|_\infty \leq \varepsilon$, we obtain

$$N \geq \frac{1}{1 - \overline{\gamma}} \log \frac{R_{\max}}{\varepsilon(1 - \gamma)}.$$

$\square$

## A.4 Proof of Theorem 4.1

*Proof.* For the maximum likelihood objective, after introducing the sequence $z_{0:T}$ of latents, we have

$$\log p(a_{0:T} \mid s_{0:T}) = \log \sum_{z_{0:T}} p(a_{0:T}, z_{0:T} \mid s_{0:T})$$
$$= \log \sum_{z_{0:T}} p(a_{0:T}, z_{0:T} \mid s_{0:T}) \frac{q(z_{0:T} \mid s_{0:T})}{q(z_{0:T} \mid s_{0:T})}$$
$$= \log \mathbb{E}_{z_{0:T} \sim q(\cdot|s_{0:T})}\left[\frac{p(a_{0:T}, z_{0:T} \mid s_{0:T})}{q(z_{0:T} \mid s_{0:T})}\right]$$
$$\geq \mathbb{E}_{z_{0:T} \sim q(\cdot|s_{0:T})}\left[\log \frac{p(a_{0:T}, z_{0:T} \mid s_{0:T})}{q(z_{0:T} \mid s_{0:T})}\right], \tag{A.1}$$

where the last inequality follows from Jensen's inequality.

From the probabilistic graphical model, we obtain that

$$q(z_{0:T} \mid s_{0:T}) = q(z_0 \mid s_{0:T}) \prod_{t=1}^T q(z_t \mid s_{0:T}, z_{0:t-1})$$
$$= q(z_0 \mid s_0) \prod_{t=1}^T q(z_t \mid s_t, z_{0:t-1}). \tag{A.2}$$

Besides, we have from the Bayes rule that

$$p(a_{0:T}, z_{0:T} \mid s_{0:T}) = p(a_0, z_0 \mid s_{0:T}) \prod_{t=1}^{T} p(z_t, a_t \mid s_{0:T}, z_{0:t-1}, a_{0:t-1})$$

$$= p(a_0, z_0 \mid s_{0:T}) \prod_{t=1}^{T} p(z_t \mid s_{0:T}, z_{0:t-1}, a_{0:t-1}) p(a_t \mid s_{0:T}, z_{0:t}, a_{0:t-1})$$

$$= p(a_0, z_0 \mid s_0) \prod_{t=1}^{T} p(z_t \mid s_t, z_{0:t-1}) p(a_t \mid s_t, z_t)$$

$$= p(z_0 \mid s_0) \prod_{t=1}^{T} p(z_t \mid s_t, z_{0:t-1}) \prod_{t=0}^{T} p(a_t \mid s_t, z_t). \tag{A.3}$$

Plugging (A.2) and (A.3) into (A.1) gives us

$$\log p(a_{0:T} \mid s_{0:T}) \geq \mathbb{E}_{z_{0:T} \sim q(\cdot \mid s_{0:T})} \left[ \log \frac{p(z_0 \mid s_0) \prod_{t=1}^{T} p(z_t \mid s_t, z_{0:t-1}) \prod_{t=0}^{T} p(a_t \mid s_t, z_t)}{q(z_0 \mid s_0) \prod_{t=1}^{T} q(z_t \mid s_t, z_{0:t-1})} \right]$$

$$= \mathbb{E}_{z_t \sim q(\cdot \mid s_t, z_{0:t-1})} \left[ \sum_{t=0}^{T} \log p(a_t \mid s_t, z_t) - \log \frac{q(z_t \mid s_t, z_{0:t-1})}{p(a_t \mid s_t, z_t)} \right]$$

$$= \mathbb{E}_{z_t \sim q(\cdot \mid s_t, z_{0:t-1})} \left[ \sum_{t=0}^{T} \log p(a_t \mid s_t, z_t) - \mathcal{D}_{\mathrm{KL}} \Big( q(z_t \mid s_t, z_{0:t-1}) \,\|\, p(z_t \mid s_t, z_{t-1}) \Big) \right],$$

where $p(z_t \mid s_t, z_{t-1})$ is the prior distribution of the latent $z_t$. $\qquad\square$

## B  STATEMENT ON THE USE OF LARGE LANGUAGE MODELS

We use LLMs only to polish the paper, such as improving clarity and grammar, without altering its substance or original ideas.

