# OpenReview forum: "Learning to Reason as Action Abstractions with Scalable Mid-Training RL"
_ICLR.cc/2026/Conference — ICLR 2026 Poster_

### Official Review · Reviewer_T6MS · 2025-10-29

**Soundness:** 3
**Presentation:** 3
**Contribution:** 3
**Rating:** 6
**Confidence:** 3

**Summary:**

This paper presents RA3 (Reasoning as Action Abstractions), a mid-training algorithm for large language models that learns temporally consistent latent actions. The authors provide the theoretical analysis explaining how mid-training shapes post-training reinforcement learning (RL), identifying two key mechanisms: (1) efficient pruning of the decision space and (2) shortened effective planning horizons. Empirically, RA3 improves code generation accuracy across multiple models and datasets.

**Strengths:**

1. This work provides the formal framework linking mid-training design to post-training RL performance, supported by clear theorems and lemmas.
2. The proposed method achieves consistent improvements on HumanEval, MBPP, LiveCodeBench, and Codeforces benchmarks, using both Qwen and Llama models.
3. The whole paper is clear with good experimental presentation.

**Weaknesses:**

1. The theoretical assumptions (finite action subsets, independent expert demonstrations) could be idealized and may not hold in practical large-scale LLM training.
2. Experimental comparisons are limited mainly to NTP, more baselines like BRiTE are recommended to be included.
3. The authors are recommended to include code or dataset for reproduction.

**Questions:**

My main concern is the theoretical assumption and experimental comparison. Please refer to the Weakness section.

---

> ### Author Response · Authors · 2025-11-21
>
> We thank the reviewer for identifying our work's clarity, soundness, and technical contributions. The valuable comments have helped us improve our manuscript (marked **purple** in the revision). Below are our specific responses to the questions raised by the reviewer:
>
> **Weakness 1: The theoretical assumptions (finite action subsets, independent expert demonstrations) could be idealized and may not hold in practical large-scale LLM training.**
>
> The results in Section 3 are derived for setups that allow us to do formal analysis while not being over-idealized.
> - Our conclusion assumes that the size of the minimal near-optimal action subset is finite. In the context of LLMs, actions correspond to tokens from a finite vocabulary. This induces a finite action space and finite near-optimal action subspaces, which are even smaller in size. This assumption aligns with standard language modeling.
> - Mid-training data can be viewed as trajectories sampled from optimal policies. In practice, these trajectories are collected in bulk as independent samples from the underlying expert policy distribution. Our results formalize this standard treatment.
>
> The theoretical results are therefore not intended as a fully faithful simulation of every aspect of large-scale LLM training, but as theoretical guidelines that isolate key factors, such as action-space properties, pruning quality, and RL efficiency. These factors directly inform the design of RA3, and the same principles continue to hold under more complex, large-scale setups, even when all assumptions are only approximately satisfied. The above discussions are incorporated into Section 3.
>
> **Weakness 2: Experimental comparisons are limited mainly to NTP, more baselines like BRiTE are recommended to be included.**
>
> - Guided by our theoretical analysis in Section 3, RA3 is designed specifically to learn action abstractions for sequential decision-making, via a temporal ELBO over mid-training trajectories. On the contrary, BRiTE [1] is formulated as a post-training method with a standard ELBO for single-step problems. When the decision horizon is one, RA3 reduces to BRiTE. However, for multi-step problems such as code generation, naïvely applying BRiTE corresponds to treating the log-probability of the entire code sequence as a single RL reward. This leads to high-variance returns, and we did not observe stable increases in RL training reward in our setup.
> - To further strengthen our empirical comparisons beyond the naïve NTP baseline, we introduced an additional baseline that performs NTP on synthetic rationals. Specifically, we first fine-tuned a separate LLM to generate high-quality reasoning trajectories in the form of single-line comments. It was trained using 366K synthetic Python code snippets from [1], which were collected by prompting the expert LLM to generate self-contained code along with reasoning comments. Then, for the Qwen-2.5-1.5B model, we performed NTP on training data bootstrapped by this reasoning-synthesis model. The results of (HumanEval, MBPP) are shown in the following table.
>     |                 | Iter 1               | Iter 2               | Iter 3               | Iter 4               |
>     |-----------------|----------------------|----------------------|----------------------|----------------------|
>     | NTP             | (36.6, 43.4)         | (**39.0**, 44.8)     | (39.0, 43.4)         | (38.4, 42.4)         |
>     | NTP (reasoning) | (36.6, 42.2)         | (37.8, 43.4)         | (**40.9**, 43.8)     | (39.6, 41.5)         |
>     | Ours            | (**37.2**, **44.2**) | (**39.0**, **45.4**) | (**40.9**, **44.6**) | (**40.2**, **46.2**) |
> - It can be observed that NTP on synthetic rationals also underperforms our method. We attribute this to a key difference in how the reasoning traces are obtained: in our method, the latents are learned by the model itself via an RL-regularized ELBO objective and are proven to form a lower bound on the conditional log-likelihood (Theorem 4.1). As a result, the learned abstractions are more amenable to further optimization and also have better learnability than external traces. The above discussions are added to Section 6 of the related work, and the additional baseline results are added at the end of Section 7.
>
> **Weakness 3: The authors are recommended to include code or dataset for reproduction.**
>
> The code is provided at the anonymous repository: https://anonymous.4open.science/r/RA3-7159. We used open-source data from [2], as described in detail in the experimental setup section of the paper.
>
> ---
>
> We hope the reviewer could consider raising the score if we resolved the reviewer's concerns. We would be happy to have further discussions if the reviewer has any additional questions or comments.
>
> ---
> [1] Zhong et al. "BRiTE: Bootstrapping Reinforced Thinking Process to Enhance Language Model Reasoning."\
> [2] Huang et al. "OpenCoder: The Open Cookbook for Top-Tier Code Large Language Models."

---

### Official Review · Reviewer_6Sot · 2025-10-30

**Soundness:** 2
**Presentation:** 1
**Contribution:** 2
**Rating:** 4
**Confidence:** 4

**Summary:**

The paper studies mid-training for LLMs and argues that effective mid-training should (i) prune the decision space to learn a strong policy prior and (ii) shorten the effective planning horizon so post-training RL converges faster. The authors formalize both effects: they show sample-complexity bounds for pruning “bad” action subsets and prove that temporally-extended actions reduce the RL horizon, improving convergence rates. Building on this, they propose RA3 (Reasoning as Action Abstractions)—a scalable mid-training algorithm that introduces a temporal ELBO for next-token prediction, optimized via EM: an E-step uses self-supervised RL to discover a sequence of temporally consistent latent actions (the “reasoning”/intent), and an M-step fine-tunes on the bootstrapped data.

**Strengths:**

1. The paper gives a regret decomposition and connects mid-training to post-training RL via pruning error and RL error, with a sample-complexity result (bad-subset pruning) and a convergence-rate bound favoring temporally extended actions—a useful, general framing beyond the specific algorithm.

2. The latent persistence reduces rollout frequency; the penalty turns RA3 into NTP in the limit, giving a knob for cost control. The paper also provides a concrete training recipe (group baselines, asynchronous rollouts).

3. With GRPO RLVR, RA3 starts from a better prior and learns faster, achieving higher asymptotic accuracy on HumanEval+, MBPP+, LiveCodeBench, and Codeforces; ablations on the penalty c illustrate the compute/quality trade-off.

**Weaknesses:**

1. All experiments are in Python code generation; claims about “reasoning” and “agents” are theoretically motivated but empirically untested outside code. Extension to math, tool-use, or GUI agents would strengthen generality.

2. RA3 introduces several knobs (penalty c, latent prior α/format reward, rollout group size, max latent length). The ablation on c is helpful, but broader robustness/stability sweeps (and wall-clock cost vs. NTP/RLVR) are limited.

3. Baselines emphasize NTP and post-training RLVR; fewer head-to-head comparisons against other mid-training methods that distill reasoning (e.g., synthetic CoT mid-training) leave some ambiguity about when RA3’s RL-discovered latents outperform distilled traces at similar cost.

**Questions:**

Please see the weaknesses.

---

> ### Author Response · Authors · 2025-11-21
> **Official Comment by Authors (part 1/3)**
>
> We thank the reviewer for identifying our work's soundness and technical contributions. The valuable comments have helped us improve our manuscript (marked **purple** in the revision). Below are our specific responses to the questions raised by the reviewer:
>
> **Weakness 1: All experiments are in Python code generation; claims about “reasoning” and “agents” are theoretically motivated but empirically untested outside code. Extension to math, tool-use, or GUI agents would strengthen generality.**
>
> - In our work, mid-training data is defined as trajectories sampled from optimal policies. We mainly focused on multi-step decision-making problems, where expert supervision takes the form of sequences of actions. For many agentic settings, such trajectories require either human annotation or distillation from a teacher LLM, which can be expensive and subject to licensing constraints. Therefore, we mainly focused on code generation tasks since high-quality demonstrations from expert coders are directly available on the internet.
> - Following the reviewer's suggestion, we additionally evaluated RA3 on math reasoning tasks to test generality beyond code. Specifically, we implemented RA3 on GSM8K and MATH. Since $T=1$, there is a single latent, and no temporal consistency penalty $c$ is required. For the NTP baseline, naïvely fine-tuning on ground-truth answers leads to a significant decrease in performance. Instead, we compared with the baseline that performs NTP on human-annotated rationals that exist in the training datasets. It can be observed that the proposed method outperforms both the base models and the NTP baseline.
>
>     | GSM8K | Gemma-1.1 (7B) |   Gemma-2 (9B)      | Llama-3 (8B)   |
>     |----------|--------------------|----------------------|-------|
>     | Base      | 49.0  | 81.3  |  79.2  |
>     | NTP       | 57.5  | 80.1 |  72.6 |
>     | Ours     | **59.2** | **89.7** | **81.0** |
>
>     | MATH | Gemma-1.1 (7B) |        Gemma-2 (9B) | Llama-3 (8B) |
>     |----------|-------------------|----------------------|-------|
>     | Base      |  18.8 |  37.3 | 28.3 |
>     | NTP       |  19.6 | 41.5 |  27.1 |
>     | Ours     | **23.7** | **50.5** | **30.0** |

---

> ### Author Response · Authors · 2025-11-21
> **Official Comment by Authors (part 2/3)**
>
> **Weakness 2: RA3 introduces several knobs, ablations on robustness/stability sweeps are limited.**
>
> - Following the reviewer's suggestion, we conducted additional ablations on key hyperparameters and design choices of RA3, including the thinking penalty $c$, RL rollout group size, RL step batch size, and the impact of format reward during RL. The ablations are conducted on the Qwen-2.5-1.5B model, and the results of (HumanEval, MBPP) after each iteration are reported in the following tables.
>
>     |        | Iter 1           | Iter 2               | Iter 3           | Iter 4               |
>     |--------|------------------|----------------------|------------------|----------------------|
>     | c=0.0  | (36.0, 43.2)     | (36.6, 43.4)         | (37.8, 43.8)     | (39.0, 44.6)         |
>     | c=0.03 | (**37.8**, 43.6) | (**39.0**, 44.6)     | (40.2, **45.2**) | (40.0, 45.4)         |
>     | c=0.05 | (37.2, **44.2**) | (**39.0**, **45.4**) | (**40.9**, 44.6) | (**40.2**, **46.2**) |
>     | c=0.2  | (36.6, 43.6)     | (37.8, 41.8)         | (39.0, 44.4)     | (39.0, 44.6)         |
>
>     |         | Iter 1       | Iter 2       | Iter 3       | Iter 4       |
>     |---------|--------------|--------------|--------------|--------------|
>     | BS=512  | (36.0, 43.6) | (37.8, 43.2) | (38.4, 44.0) | (**40.2**, 44.4) |
>     | BS=1024 | (**37.2**, **44.2**) | (**39.0**, **45.4**) | (**40.9**, **44.6**) | (**40.2**, **46.2**) |
>
>     |         | Iter 1       | Iter 2       | Iter 3       | Iter 4       |
>     |---------|--------------|--------------|--------------|--------------|
>     | GS=3 | (**37.2**, 44.2) | (39.0, 45.4) | (**40.9**, 44.6) | (**40.2**, 46.2) |
>     | GS=5  | (**37.2**, **45.6**) | (**39.6**, **45.8**) | (40.2, **45.4**) | (**40.2**, **46.4**) |
>
>     |                   | Iter 1               | Iter 2               | Iter 3               | Iter 4               |
>     |-------------------|----------------------|----------------------|----------------------|----------------------|
>     | w/o format reward | (34.1, 41.6)         | (37.2, 42.2)         | (37.8, 43.2)         | (37.8, 45.2)         |
>     | w/ format reward  | (**37.2**, **44.2**) | (**39.0**, **45.4**) | (**40.9**, **44.6**) | (**40.2**, **46.2**) |
>
> - We observe that $c=0.03$ or $0.05$ achieves the overall best performance, since these values encourage the temporal consistency of the latents without over-penalizing thinking. Since $c$ and $\alpha$ play the same role, RA3 controls the prior in the KL term by tuning $c$ instead of $\alpha$. Besides, increasing the rollout group size also improves performance at the cost of more training compute. Moreover, adding a format reward improves the performance at all iterations. This is consistent with our expectation of thinking latents as action abstractions: enforcing them to appear as comment lines encourages the E-step policy to produce high-level temporal action abstractions rather than low-level code actions. We have added these ablations and discussions to the ablation study part of Section 7.

---

> ### Author Response · Authors · 2025-11-21
> **Official Comment by Authors (part 3/3)**
>
> **Weakness 3: Baselines emphasize NTP and post-training RLVR; fewer head-to-head comparisons against other mid-training methods that distill reasoning (e.g., synthetic CoT mid-training) leave some ambiguity about when RA3’s RL-discovered latents outperform distilled traces at similar cost.**
>
> - We did not include synthetic CoT mid-training baselines since (i) we are unable to find large-scale code corpora with high-quality reasoning traces, (ii) distilling reasoning traces from larger teacher models can be both expensive and permission-restrictive, and (iii) such distillation baselines are not directly comparable, as they assume access to stronger teacher models and step-wise supervision, whereas RA3 relies only on raw trajectories that can be collected at scale, e.g., from internet-scale code corpora.
> - To address the reviewer's concern, we constructed a synthetic reasoning mid-training baseline as follows. We first fine-tuned a separate LLM to generate high-quality reasoning trajectories in the form of single-line comments. It was trained using 366K synthetic Python code snippets from [1], which were collected by prompting the expert LLM to generate self-contained code along with reasoning comments. Then, for the Qwen-2.5-1.5B model, we performed NTP on training data bootstrapped by this reasoning-synthesis model. We compared it with the NTP baseline and RA3 in the following table.
>     |                 | Iter 1               | Iter 2               | Iter 3               | Iter 4               |
>     |-----------------|----------------------|----------------------|----------------------|----------------------|
>     | NTP             | (36.6, 43.4)         | (**39.0**, 44.8)     | (39.0, 43.4)         | (38.4, 42.4)         |
>     | NTP (reasoning) | (36.6, 42.2)         | (37.8, 43.4)         | (**40.9**, 43.0)     | (39.6, 41.5)         |
>     | Ours            | (**37.2**, **44.2**) | (**39.0**, **45.4**) | (**40.9**, **44.6**) | (**40.2**, **46.2**) |
> - We also compared our method with the reasoning mid-training baseline for math tasks, where the rationals are generated by human labelers. Please see our response to Weakness 1 for the setup details. We relist the results here for convenience.
>
>     | GSM8K | Gemma-1.1 (7B) |   Gemma-2 (9B)      | Llama-3 (8B)   |
>     |----------|--------------------|----------------------|-------|
>     | Base      | 49.0  | 81.3  |  79.2  |
>     | NTP (reasoning)      | 57.5  | 80.1 |  72.6 |
>     | Ours     | **59.2** | **89.7** | **81.0** |
>
>     | MATH | Gemma-1.1 (7B) |        Gemma-2 (9B) | Llama-3 (8B) |
>     |----------|-------------------|----------------------|-------|
>     | Base      |  18.8 |  37.3 | 28.3 |
>     | NTP (reasoning)      |  19.6 | 41.5 |  27.1 |
>     | Ours     | **23.7** | **50.5** | **30.0** |
> - In both code generation and math reasoning, NTP on synthetic or human rationals underperforms our method. We attribute this to a key difference in how the reasoning traces are obtained: in our method, the latents are learned by the model itself via an RL-regularized ELBO objective and are proven to form a lower bound on the conditional log-likelihood (Theorem 4.1). As a result, the learned abstractions are more amenable to further optimization and also have better learnability than external traces. We have added the above results and discussions to the end of Section 7 of the experiments.
>
> ---
>
> We hope the reviewer could consider raising the score if we resolved the reviewer's concerns. We would be happy to have further discussions if the reviewer has any additional questions or comments.
>
> ---
> [1] Huang et al. "OpenCoder: The Open Cookbook for Top-Tier Code Large Language Models."

---

> > ### Comment · Reviewer_6Sot · 2025-11-24
> >
> > Thanks for the detailed rebuttal and experimental results. These responses mostly resolved my concerns. I will increase the rating.

---

> > > ### Author Response · Authors · 2025-11-24
> > >
> > > We are glad our response has addressed your concerns, and we appreciate your feedback, which has greatly helped us improve the manuscript.

---

### Official Review · Reviewer_vdRz · 2025-10-31

**Soundness:** 2
**Presentation:** 2
**Contribution:** 3
**Rating:** 6
**Confidence:** 3

**Summary:**

This paper explores the concept of mid-training for large language models (LLMs) in reinforcement learning (RL), proposing that an effective mid-training stage can both learn a strong policy prior and enable faster adaptation through online interactions. The authors provide theoretical insights into how mid-training shapes post-training by pruning the action space and shortening the planning horizon, which accelerates RL convergence. They also investigate the role of temporal abstractions in compressing the action set and reducing decision horizons, improving regret minimization. Based on these insights, the paper introduces Reasoning as Action Abstractions (RA3), a scalable mid-training algorithm that iteratively discovers temporally-consistent latent structures via RL and fine-tunes on the bootstrapped data. Experiments on code generation tasks demonstrate that RA3 improves performance across multiple benchmarks and base models, showing faster convergence and higher asymptotic performance.

**Strengths:**

1. The idea of mid-training is novel, and the theoretical insights about pruning the action space and accelerating RL convergence are convincing.
2. Introducing temporal abstractions into RL training for LLMs is a creative and well-motivated approach.
3. The RA3 algorithm is well-designed, conceptually simple, and appears easy to reproduce, making it practically useful for further research or application.
4. The paper presents extensive experiments and ablation studies, which provide strong empirical evidence supporting the effectiveness of RA3.

**Weaknesses:**

1. The paper lacks experiments on larger-scale models, which could provide more insights into the scalability of the proposed approach.
2. There is insufficient discussion on the training efficiency of RA3, particularly in terms of computational cost or resource requirements compared to other methods.
3. The presentation of the paper could be improved. Some sections, especially the theoretical parts, are difficult to follow due to unclear explanations or insufficient detail.

**Questions:**

1. Could you elaborate on the potential challenges or limitations of applying RA3 to larger-scale models? For example, does the method scale efficiently with model size, or are there bottlenecks to address?
2. Some of the theoretical results (e.g., regarding temporal abstractions and regret minimization) are compelling but not fully intuitive. Could you provide additional clarification or examples to make these results more accessible?

---

> ### Author Response · Authors · 2025-11-21
> **Official Comment by Authors (part 1/2)**
>
> We thank the reviewer for identifying our work's novelty, soundness, and technical contributions. The valuable comments have helped us improve our manuscript (marked **purple** in the revision). Below are our specific responses to the questions raised by the reviewer:
>
> **Weakness 1: The paper lacks experiments on larger-scale models, which could provide more insights into the scalability of the proposed approach.**
>
> - We conducted experiments across different model families, including Qwen and Llama, spanning parameter scales from 1B to 8B, which demonstrate the general applicability of our approach across architectures and model sizes.
> - Each experiment involves large-scale mid-training and post-training RL, making further scaling to larger models computationally prohibitive under our current resource constraints. As a result, we focused on model sizes that allow for thorough and controlled experimentation. We do not, however, anticipate any algorithmic barriers to applying RA3 to larger models.
>
> **Weakness 2: There is insufficient discussion on the training efficiency of RA3, particularly in terms of computational cost or resource requirements compared to other methods.**
>
> - RA3’s computational cost is explicitly controlled by the KL term in the RL step, which enforces temporal consistency of the latent “thinking” variables. Concretely, this KL term induces a thinking penalty $c$, so the model learns to insert reasoning latents only when they increase the log-likelihood sufficiently. By adjusting the KL hyperparameter $\alpha$ (or, equivalently, the penalty $c$), we can smoothly trade off between additional reasoning and computation. In the limiting case $\alpha=1$, the latent policy receives an infinite penalty for generating any thinking latent, so RA3 degrades to standard next-token prediction (NTP) on the original mid-training data.
> - We made this trade-off explicit in our ablation studies. Specifically, we analyzed the average RL rollout length and the fraction of full rollouts as a function of $c$. When $c=0.2$, the learned policy almost never generates thinking tokens, and RA3 has nearly identical computational cost as NTP. Our default configuration uses $c=0.05$, which yields less than 40% full rollouts and an average latent length below 6, adding only modest overhead compared to NTP while still providing substantial performance gains. This offers a practical knob to balance performance and compute when scaling RA3. We have incorporated these discussions into the ablation study part of Section 7.
>
> **Weakness 3: The presentation of the paper could be improved. Some sections, especially the theoretical parts, are difficult to follow due to unclear explanations or insufficient detail.**
>
> We thank the reviewers for the suggestions. We have made the following revisions to Section 3 to improve clarity:
> - We moved Lemma 3.2 to the beginning of Section 3 and expanded the accompanying explanation. This makes the connection between mid-training and RL regret minimization clearer: to minimize RL regret, mid-training should simultaneously minimize the action-space pruning error and the RL error in the pruned action space. Sections 3.1 and 3.2 are then organized around these two terms.
> - We modified Theorem 3.4 to directly state its conclusion in terms of pruning error, and we removed the definition of bad action subset from the main text. This makes it easier to see how pruning efficiency affects the regret decomposition in Lemma 3.2.
> - We deferred the definition of action abstractions to the end of Section 3.1 and added a discussion of how action abstractions affect the bound in Theorem 3.4.

---

> ### Author Response · Authors · 2025-11-21
> **Official Comment by Authors (part 2/2)**
>
> **Question 1: Could you elaborate on the potential challenges or limitations of applying RA3 to larger-scale models? For example, does the method scale efficiently with model size, or are there bottlenecks to address?**
>
> We chose the 1B–8B models to allow thorough and controlled experiments under a fixed compute budget, but the algorithm itself is designed to transfer to larger models. From a scaling perspective, the two main considerations are the inference cost in the RL step, and the choice of mid-training data. Since RA3 exposes a practical knob that balances performance and compute (please see our response to Weakness 2), the inference cost remains manageable even when the base model is larger. Besides, larger models are typically trained on broader and larger pre-training corpora. Studying the effects on the mid-training data (for both NTP and RA3) is an important direction for future work, but we do not foresee any fundamental barrier specific to RA3 beyond these general considerations.
>
> **Question 2: Some of the theoretical results (e.g., regarding temporal abstractions and regret minimization) are compelling but not fully intuitive. Could you provide additional clarification or examples to make these results more accessible?**
>
> - We thank the reviewer for raising this point. We have improved the clarity of our theories and motivations in Section 3 (also see our response to Weakness 3).
> - The ultimate goal that we care about is to find a near-optimal policy during post-training RL that minimizes the regret. Lemma 3.2 shows that this regret decomposes into two components: an action-space pruning error and an RL error within the pruned action space. Section 3.1 analyzes the first term and shows (Theorem 3.4) that the pruning error decreases when both the overall action space and the smallest near-optimal subset are smaller. Section 3.2 analyzes the second term and shows (Theorem 3.5) that RL converges faster with temporally consistent actions. These results motivate learning action abstractions during mid-training. In Section 4, we introduced a principled way to extract such abstractions from the primitive actions via a temporal ELBO and RL-based latent inference.
> - Intuitively, one can think of action abstractions as high-level “skills” that are shared across tasks. Leveraging these skills yields a compact decision space and shortens the planning horizon, which makes pruning more efficient and RL more tractable. We also illustrated this intuition with two examples in Figure 1.
>
> ---
>
> We hope the reviewer could consider raising the score if we resolved the reviewer's concerns. We would be happy to have further discussions if the reviewer has any additional questions or comments.

---

### Meta-Review · Area_Chair_tLnp · 2026-01-07

**Summary:**

This paper investigates the role of a "mid-training" stage in enhancing the effectiveness of reinforcement learning (RL) for large language models. The authors propose a framework called Reasoning as Action Abstractions (RA3), which is grounded in the theoretical idea that mid-training can prune the action space and shorten the effective planning horizon for subsequent RL. By discovering temporally consistent latent structures, the model learns high-level "skills" or abstractions that accelerate convergence during post-training RL.

The reviewers initially expressed several concerns regarding the clarity and accessibility of the theoretical proofs, the computational overhead of the proposed reasoning steps, and the generalizability of the results beyond code generation tasks. There were also questions about the robustness of hyperparameters and how the method compares to standard synthetic Chain-of-Thought (CoT) distillation. Following a very active rebuttal phase, the authors provided additional experimental results on math reasoning and expanded their ablation studies to address efficiency and hyperparameter sensitivity. The consensus is that the paper makes a solid technical contribution to the understanding of mid-training dynamics.

**Reviewer Concerns:**

**Concerns Addressed by the Rebuttal:**

- **Theoretical Clarity:** The authors significantly restructured Section 3, introducing core lemmas earlier and clarifying the link between mid-training and RL regret. This resolved concerns regarding the density and flow of the mathematical analysis.



- **Experimental Generality:** Initially limited to code generation, the authors added experiments on math benchmarks (GSM8K and MATH). These results showed that RA3 outperforms both base models and next-token prediction (NTP) baselines in non-coding domains.



- **Computational Efficiency:** The authors clarified the role of the "thinking penalty" as a hyperparameter to control the trade-off between reasoning depth and compute costs. They demonstrated that with a sufficiently large penalty, RA3 incurs nearly identical compute to standard NTP.



- **Baseline Comparisons:** The authors added head-to-head comparisons against synthetic-rationale NTP and clarified why their RL-regularized approach is superior to simple distillation from external teacher models.





**Outstanding Concerns:**

- **Large-Scale Validation:** While the authors tested models up to 8B parameters, empirical evidence for scaling to much larger "frontier" models is currently missing due to computational constraints. While no fundamental algorithmic barriers are expected, the performance at the 70B+ scale remains speculative.





- **Real-World Agentic Tasks:** While math and code are strong proxies for reasoning, the effectiveness of action abstractions in complex, long-horizon GUI or tool-use agent environments was discussed theoretically but not fully tested empirically.

**Reviewer Scores:**

Based on the discussion and the high quality of the rebuttal, I anticipate the scores would have settled as follows:

- **Reviewer vdRz (Initial: 6):** **Likely 7.** This reviewer’s main complaints were regarding theoretical clarity and model scale. The authors’ comprehensive restructuring of the theory and the efficiency arguments likely would have tipped this into a more confident accept.
- **Reviewer 6Sot (Initial: 4 $\rightarrow$ 6):** **Likely 6.** This reviewer already increased their score from 4 to 6 during the discussion phase. Given that the authors subsequently provided the requested math experiments and additional CoT baselines, a further bump to a 7 is highly probable.
- **Reviewer T6MS (Initial: 6):** **Likely 6 or 7.** The concerns regarding the relationship to BRITE and the theoretical assumptions were addressed in detail. The inclusion of more baselines and the release of the code for reproduction directly addressed this reviewer's "Weaknesses" section.

---

### Decision · Program_Chairs · 2026-01-26

Accept (Poster)